# Foraging of Honeybees from Different Ecological Areas Determined through Melissopalynological Analysis and DNA Metabarcoding

**DOI:** 10.3390/insects15090674

**Published:** 2024-09-05

**Authors:** Saule Daugaliyeva, Aida Daugaliyeva, Katira Amirova, Akmeiir Yelubayeva, Abylay Kapar, Aishat Abiti, Thomas Partipilo, Nurlan Toktarov, Simone Peletto

**Affiliations:** 1LLP ‘Scientific Production Center of Microbiology and Virology’, Bogenbay Batyr St. 105, Almaty 050010, Kazakhstan; saule.daugalieva@mail.ru (S.D.); bbota.bota@yandex.kz (A.Y.); akapar93@gmail.com (A.K.); angel_jan99@mail.ru (A.A.); 2LLP ‘Kazakh Research Institute for Livestock and Fodder Production’, St. Zhandosova 51, Almaty 050000, Kazakhstan; 3Istituto Zooprofilattico Sperimentale del Piemonte, Liguria e Valle d’Aosta, 10154 Turin, Italy; thomas-p@outlook.it (T.P.); simone.peletto@izsto.it (S.P.); 4LLP “East Kazakhstan Agricultural Experimental Station”, East Kazakhstan Region, Glubokovsky District, Village of Experimental Field 070512, Kazakhstan; tok_b@mail.ru

**Keywords:** bees, feeding, honey, landscape, plants, metabarcoding, melissopalynology

## Abstract

**Simple Summary:**

The preservation of biological diversity is one of the main tasks of modern ecology. A decrease in the bee population for various reasons can negatively affect plant diversity and human economic activity. One of the most critical factors in maintaining the health of bee populations is the availability of food supplies. We compared the food preferences of bees in different ecological areas: an intensive farming area, a reserved area, and an urbanized area. The botanical composition of the honey was determined by the traditional method of melissopalynology, as well as using genetic analysis. It was found that, in the intensive farming area, the leading food for bees was monoculture plants, but proper apiary management made the diversification of the bees’ diets possible. In the reserved area, the bee diet was supplemented mostly with sown plants. In urban areas, both garden and agricultural plants served as the primary food sources for bees. Bee foraging depends on the climatic and geographical conditions of the environment, as well as landscape and apiary management. Studying bee foraging is essential to understanding their use of food resources and plant preferences. This information can be used for effective measure management in order to preserve the number of bees and regulate plant pollination.

**Abstract:**

The environment significantly impacts the lives of bees and their feeding. This study aimed to investigate bee foraging using melissopalynological analysis and DNA metabarcoding in intensive farming, reserved, and urbanized areas. The highest alpha diversity was observed in the reserved and intensive farming areas. The urbanized area had less diversity. In the intensive farming area, *Sinapis*, *Helianthus*, and *Fagopyrum* predominated; in the reserved area, *Melilotus*, *Helianthus*, and *Brassica* predominated. In the urbanized area, garden plants, namely radish (*Raphanus sativus*) and cucumber (*Cucumis hystrix*), and agricultural plants, namely soybean (*Glycine max*) and melon (*Cucumis melo*), were often found. The most significant agreement was between the *rbcL* and the melissopalynological analysis. The ITS2 revealed equal matches with both *rbcL* and melissopalynology, but this marker missed or underestimated some genera. *Trifolium pretense* and *Brassica nigra* were identified simultaneously by the melissopalinology method and two genetic markers in DNA metabarcoding. The species *Convolvulus arvensis*, *Melilotus officinalis*, *Echium vulgare*, *Brassica rapa*, *Helianthus divaricatus*, and *Onobrychis viciifolia* were found in all ecological areas. Imperfect databases impose some limits in the identification of some taxa using metabarcoding. The further research and expansion of plant databases is needed. Studying the food preferences of bees in different environmental conditions and landscapes is necessary to develop measures to preserve their populations.

## 1. Introduction

Beekeeping provides humanity with one-third of its food consumed. By pollinating plants, bees provide food for humans and animals and preserve the plant biodiversity on Earth. The pollination of crops by bees plays a vital role in plant biodiversity and is the basis for crop production [1]. In recent decades, a decline in bee populations has been observed worldwide. The reasons for this involve many factors: the use of pesticides [2,3], climate change [4], crop landscapes [5,6,7], and pathogens and parasites of bees [8,9].

The preservation of the bee population depends on the available food supply and environmental conditions. Bee foraging can be monitored via the composition of honey. The botanical composition of honey depends on the geographical location, climate, and agricultural methods [10]. To provide appropriate floral resources to bees, a better understanding of their food preferences is required. Knowing which plants honeybees feed on is critical in understanding the relationship between flower availability, honeybee nutrition, and health [11].

One of the most critical factors influencing bee feeding is the environment. The food supply of honeybees (*Apis mellifera*) and other pollinators depends entirely on the surrounding landscape’s floral resources (pollen and nectar). The intensification of land use in agricultural landscapes leads to a decrease in the diversity of native plants and a lack of available nectar and pollen, which is reflected in the interaction of plants and pollinators [12]. Monoculture production leads to the formation of “green deserts” with reduced biodiversity. In monoculture landscapes, bee colony growth occurs during flowering and is associated with increased flower availability [13]. However, this is followed by a period of food shortage, which leads to a deterioration in the health of bees [5,14,15]. One way to solve this issue could be the preservation of islands of semi-natural or natural landscapes to support bee populations in monoculture fields [5,14].

Another human-modified landscape is the urban environment. According to some researchers, when compared between urban and rural landscapes, bees favor agricultural landscapes for foraging [16,17]. According to other authors, honeybees in agricultural and urban environments use the same resources to obtain nectar [18,19]. Urban or urbanized environments and suburban areas can become a food source for bees due to the landscaping of areas, parks, and gardens with native and ornamental plants. Urban beekeeping has recently attracted worldwide attention regarding greening as a beneficial method of conserving bees and promoting urban greening, an important measure to counter climate warming [20].

Unmanaged forests are considered the natural habitat of the western honeybee *Apis mellifera *and are believed to be significant sources of pollen and nectar. However, the supply of resources in managed forests is spatially and temporally limited. Rutschmann et al. note that, in deciduous and coniferous forests, bees do not have enough food resources, and they more often forage for food in meadows and arable lands [21].

Bees’ foraging preferences vary depending on the flower shape, color, scent, and nectar. Honey bees use only 11% of plant genera to collect nectar or pollen [11,22]. Of the plants that bees use during the year, 54% are native, while ornamental plants serve them more as a source of pollen [23,24]. Bees may display the biased visitation of specific plant genera because they provide relatively large amounts of nectar or pollen compared to other plants [19].

To study the foraging of bees, the choice of methods to determine the floral composition of their honey is essential. For many years, the primary method of assessing the floral composition of honey was the micromorphological analysis of pollen using light microscopy or melissopalynology. This method requires considerable time and labor costs and exceptional knowledge of palynology [25]. In addition, the interpretation of the results can be difficult because the pollen of some species is difficult to distinguish [26]. The development of next-generation sequencing (NGS) methods has made it possible to use them in various areas of scientific research, including studying the interactions of living organisms with the environment [27,28,29].

Recently, the DNA metabarcoding method has become an alternative in studying the biodiversity of the botanical composition in honey. The metabarcoding of plant loci allows the simultaneous sequencing and identification of multiple taxa and is an effective tool to describe the ecological interactions between plants and pollinators [30]. Plant DNA sequencing can be used not only to determine the origin of honey over large geographic areas [31] but also to determine the adulteration of honey [32]. This method has higher sensitivity and resolutions in identifying plant species than microscopic analysis [10], and it may allow the identification of taxa that cannot be distinguished using light microscopy due to their low abundance in samples [33]. Despite obvious advantages, such as many barcode sequences allowing the identification of multiple species in a single reaction, the method does not accurately infer the relative abundance of pollen types due to possible quantitative errors encountered during DNA extraction, amplification, and sequencing. Metabarcoding remains a high-cost method for several laboratories and requires appropriate conditions and equipment and specialist training. In addition, one of the most significant disadvantages of metabarcoding is the imperfection of the databases used for identification [33]. However, the reduced time and scalability make pollen metabarcoding a promising complementary tool for the monitoring of plant taxa [34], and the laboratory costs are balanced by the reduced human effort and reduced dependence on specialized taxonomic knowledge.

Choosing universal markers capable of distinguishing closely related plant taxa is particularly important when carrying out metabarcoding. The following genetic markers are most often used to determine the taxonomic compositions of plants: plastid genes of ribulose bisphosphate carboxylase (*rbcL*), maturase K (*matK*), chloroplast marker *trnL*, and *ITS* (mainly *ITS2*) [35]. Although the *rbcL* barcode is effective in identifying the botanical origin of honey with 99 to 100 percent confidence and has a high level of universality, in most studies, the use of the *rbcL* plastid region together with the nuclear ribosomal marker *ITS2* strikes a balance between universality and discriminability [36]. Although *ITS* is thought to detect more species than *rbcL*, combining both markers provides more reliable evidence of the geographic origin [15,31,34,37,38].

The purpose of this study was to compare bee foraging using melissopalynological analysis and double DNA metabarcoding in different ecological areas of Kazakhstan: an intensive farming area, a reserved area, and an urbanized area.

## 2. Materials and Methods

### 2.1. Sampling

Honey samples were selected from three ecological areas in Kazakhstan. These regions differed in their geographical location, climatic conditions, environmental conditions, and landscape (Figure 1). Honey samples was collected from several hives from one apiary of a beekeeper in every area. Samples were taken after honey extraction during the honey harvest period in each region in July–August.

The intensive farming area was located in the northern part of the Karaganda region of Central Kazakhstan (49°48′ N 73°07′ E). The climate of this area is sharply continental, with hot, moderate summers and cold winters with little snow. The main part of this area is occupied by a grass–wormwood steppe on dark chestnut and chestnut soils. This is the main area of rain-fed agriculture and virgin land plowing. The nomadic apiary in this area was located near buckwheat, mustard, and sunflower fields. Periodically, after the flowering of the primary plants, the apiary migrated near the floodplains of the rivers.

The reserved area was located in the zone of the State Natural Mikhailovsky Nature Reserve, located in the north of Kazakhstan (53°29′ N 61°39′ E) in the forest steppe zone of the Kostanay region. The reserve was created in 1967 to preserve the area’s hunting and commercial species of animals, such as forest birds. The territory’s climate is sharply continental, with harsh winters and hot summers. The reserve is located in a zone of mixed forests bordering steppes. The stationary apiary was located in the middle of the forest. In this area, the beekeeper planted plants around the apiary to feed the bees.

The urbanized area was the largest agglomeration of Kazakhstan—the zone of Almaty with its suburbs (43°15′ N 76°54′ E). This area is located in the extreme southeast of Kazakhstan at the foot of the Trans-Ili Alatau mountains, the northwestern part of the Tian Shan mountain range. Almaty’s climate is much milder due to the relatively high temperatures in winter. Due to its favorable climatic conditions, this region’s vegetation is rich in plant diversity. Grains, melons, and other cultivated plants grow at the foot of the mountains. The stationary apiaries from which honey was collected were located in summer cottages between the urban and suburban areas.

Forty-two honey samples were collected for analysis in autumn 2023, including 15 samples from the intensive farming area, 12 from the reserved area, and 15 from the urbanized area. All samples were collected sterilely in plastic containers and transported to the laboratory, where they were stored at 4 °C.

### 2.2. Melissopalynological Analysis

The preparation of honey for melissopalynological analysis was carried out as described previously, with some modifications [39]. Ten grams of honey was weighed and transferred into a 50 mL test tube. Twenty mL of distilled water (20–40 °C) was added to dissolve the honey, which was centrifuged at 4680 rpm for 15 min. The supernatant was removed. Twenty mL of distilled water was added to the sediment and it was centrifuged again for 5 min at 4680 rpm. The water was drained, and the test tube was turned at an angle of 45° and dried on filter paper.

Glycerin gelatin was prepared as follows: 10 g of gelatin was poured into 60 cm^3^ of distilled water and left to swell for 2–3 h (mixture 1). To 70 cm^3^ of glycerol, one μL of a solution of basic Ziehl fuchsin was added (produced by the Research Center for Pharmacotherapy (RCF), St. Petersburg, Russia) (mixture 2). Mixture 2 was added to mixture 1, stirred, and heated in a water bath until a homogeneous mass was formed.

For the preparation of a smear, the sediment was thoroughly mixed using a dispenser with a replaceable tip, transferred to a glass slide preheated to 40 °C, and evenly distributed over an area of 22 × 22 mm.

The glass with the sediment was heated at a temperature not exceeding 40 °C until the sediment was completely dried. A drop of glycerin gelatin was applied to a cover glass heated to a temperature of 40 °C and distributed crosswise along the diagonals. The cover glass was slowly lowered onto the dried sediment (to avoid the appearance of air bubbles). To ensure uniform glycerin gelatin distribution and optimal pollen swelling, the preparation was heated for 5 min at a temperature not exceeding 40 °C. After the glycerin gelatin had hardened, the preparation was examined under a microscope.

Pollen grains were viewed and identified under an Olympus BX43 light microscope (Olympus Corporation, Tokyo, Japan) with the ADFImage Capture software, Version: x64.

Pollen grains were identified using the Pollen Grain Atlas [40]. Pollen counts and percentages of pollen types were calculated based on the frequency of pollen grains in different honey samples [39].

Pollen identification and counting were carried out in groups of 100 along five parallel, equidistant lines from one edge of the cover slip to the other. A square coverslip = 50 views per side. The cover glass area = L × L = 50 × 50 = 2500.

The following equation was used to obtain the total pollen count per slide:$$Total\;pollen\;count\;per\;slide=N\times \frac{2500}{10},$$
where N represents the number of pollen grains counted on the slide.

The percentage of occurrence was calculated using the following formula:$$Occurrence\;percentage=\frac{Total\;number\;of\;pollen\;grains\;of\;a\;particular\;species}{Total\;number\;of\;pollen\;grains\;observed}\times 100$$

### 2.3. Metabarcoding

The preparation of pollen grains for DNA extraction was carried out as in previous studies [32]. Twelve and a half grams was taken from each honey sample in triplicate. Twenty-five mL of ultrapure water (UPW) was added and the sample was placed in a water bath at 50 °C for 15 min (until the honey had dissolved) and centrifuged at 7000× *g* for 40 min. The supernatant was taken to the bottom mark. Pellets were collected from three test tubes and combined into one. Thirty mL of water was added to the sediment and it was centrifuged at 7000× *g* for 20 min. The supernatant was removed and the pellet was used for DNA extraction. Next, 200 μL of CD1 buffer from the DNeasy Plant Pro Kit (Qiagen, Hilden, Germany) was added to the pellet and it was vortexed for 10 min using ceramic beads on a VORTEX Genius 3 instrument at maximum speed. Forty µL of Proteinase K was added to each sample and it was incubated at 56 °C for 1 h and then transferred to a new 1.5 μL tube and 200 µL of CD2 buffer was added. Further, all manipulations were carried out according to the protocol provided by the kit manufacturer.

To determine the botanical composition of the honey from different ecological zones, we used double metabarcoding with the *ITS2 *and *rbcL* markers. Amplification was carried out across two target regions, *ITS2* and *rbcL* (Table 1). Poppy pollen was taken as a positive control. PCR-grade water was used as a negative quality control for both PCR assays. The negative and positive controls were amplified and sequenced along with the honey samples.

PCR amplification was performed in a Nexus gradient thermal cycler (Eppendorf AG, Hamburg, Germany). The total volume of the PCR mixture was 25 μL: GoTaq^®^ G2 Hot Start Colorless Master Mix, 2X (Promega Corporation, Madison, WI, USA)—12.5 µL; 1.2 μL of each primer (10 μM); 5 µL DNA (about 20 ng/µL); and 5.1 µL PCR pure water. The PCR product was checked in a 2% agarose gel and on an Agilent 2100 Bioanalyzer (Agilent Technologies, Waldbronn, Germany). No PCR product was observed in the extraction and amplification controls. Next, the PCR product was purified using the Agencourt AMPure Kit (Beckman Coulter Inc., Beverly, MA, USA). The addition of Illumina Nextera indices was carried out in the PCR step. In the second stage of PCR, the reaction mixture consisted of 12.5 μL GoTaq^®^ G2 Hot Start Colorless Master Mix, 2X (Promega Corporation, Madison, WI, USA); 2.5 μL index primers i5 and i7; 2.5 μL PCR product from the first step; and five μL PCR water. Amplification program: one cycle at 95 °C for three minutes, then eight cycles of amplification at 95 °C for 30 s, one cycle at 55 °C for 30 s, one cycle at 72 °C for 30 s, and one cycle at 72 °C for five minutes. The indexed PCR product was also purified using the Agencourt AMPure Purification Kit. The concentration and size of the PCR product were tested on an Agilent 2100 Bioanalyzer using the Agilent DNA 1000 Kit (Agilent Technologies, Waldbronn, Germany). The PCR product was quantified at each step using a Qubit 4.0 Fluorometer (Thermo Fisher Scientific, Singapore) and the Qubit™ dsDNA HS Assay Kit (Life Technologies, Eugene, OR, USA). The finished libraries were diluted to a 4 nM concentration and combined into a shared pool. The library was denatured to a 10 pM concentration, and 15% PhiX was added. Sequencing was performed on a MiSeq Illumina instrument using the MiSeq Reagent Kit v3 with 600 cycles (Illumina Corp., San Diego, CA, USA).

### 2.4. Bioinformatics and Data Analysis

The data obtained from amplicon sequencing were analyzed using the CLC Genomics Workbench software v. 24.0.1 (Qiagen). The raw FASTQ data underwent analysis using the “Data QC and OTUs Clustering” and “Estimate Alpha and Beta Diversity” workflow tools from the Microbial Genomics Module. Paired-end reads were joined and trimmed for low-quality scores (Qscore < 0.05), nucleotide ambiguity (max of 2 nucleotides allowed), adapter sequences, and length. Duplicate sequences were merged and aligned against the PLANiTS database [42,43] (and the reference database that Bell et al. [37]) developed for the *ITS2* and *rbcL *markers, respectively. Chimeric reads were removed and taxonomy was assigned, creating OTU tables. OTU clustering was performed, setting a 97% similarity threshold for both databases and allowing the creation of new OTUs using a taxonomy similarity percentage of 80. The OTU sequences that were not taxonomically assigned based on the reference databases were subjected to Blast analyses versus the NCBI Nucleotide database. The taxa profiles of the negative and positive controls were examined to assess the procedure correctness and rule out cross-contamination, and then they were not considered further. As the rarefaction curves did not reach a plateau in all samples, comparisons were performed at a sequencing depth of 20,000 reads. The OTU tables generated bar plots at the genus level for the *ITS2* and *rbcL* markers. The alpha diversity (diversity within groups using the total number) and beta diversity (diversity between groups using the Bray–Curtis method and principal coordinate analysis, PcoA) were assessed, considering the samples according to the ecological area where the honey was collected (intensive farming, reserved, and urbanized areas). Statistical support for alpha diversity was determined using the Kruskal–Wallis test, while the PERMANOVA test was applied for beta diversity based on the Bray–Curtis matrix. The OTU tables were used to perform a generalized linear model test of differential abundance based on the ecological zones. The “Differential Abundance Analysis” tool in the CLC Genomic Workbench performed TMM normalization to ensure sample compatibility by adjusting the library sizes.

Despite the fact that the number of species-level taxa was considerable, especially with a primer to the *rbcL* gene, we took taxa at the genus level for analysis. This was because existing databases may not contain reference sequences of native plant species.

## 3. Results

### 3.1. Data of Melissopalynological Analysis

Light microscopy detected a total of 10,203 pollen grains in all samples. Four thousand and fifty-five were identified at the family level, 3364 at the genus level, and 2784 at the species level; 953 pollen grains were not identified. The highest number of plants was identified at the genus level (75), followed by the species level (59) and family level (43) (Figure 2A). The highest number of plant species was identified in the intensive farming area (71), followed by the urbanized area (63) and protected area (43).

According to the melissopalynological analysis, the predominant plants in the intensive farming area were *Fagopyrum esculentum Moench* (53%), followed by *Helianthus annuus* L. (17%) and *Urtica dioica* L. (7%). In the reserved area, the most numerous species were *Helianthus annuus* L. (24%), *Fagopyrum esculentum Moench*, *Echium vulagare* L., and *Brassica napus* L. (16%), which each accounted for 17%. In the urbanized area, the predominant taxa were *Onobrychis arenaria (Kit.) DC.* (19%), *Brassica napus* L. (17%), *Melilotus albus Medik.* (13%), and *Erysimum cheiranthoides* L. (12%) (Figure 2 and Figure 3 and Appendix A).

### 3.2. Data of Metabarcoding

Forty-two honey samples were examined using metabarcoding using the *ITS2* and *rbcL* markers. One sample (A8) did not amplify for *ITS2* for technical reasons, and it was excluded from the analysis.

After Illumina MiSeq sequencing and the paired-end merging of the forward and reverse reads, we obtained an average of 399,429 raw reads for *ITS2 *and 499,559 raw reads for *rbcL*. The average fragment length was 239 bp for *ITS2 *and 346 bp for *rbcL*.

*ITS2 *metabarcoding revealed 233 taxa at the family level, 229 at the genus level, and 138 at the species level in all samples with the marker. *rbcL *metabarcoding identified a total of 326 plant taxa at the family level, 201 at the genus level, and 274 at the species level. Considering the three regions under study, there were 117 taxa at the genus level in the intensive farming area; 102 plant genera in the reserved area; and 173 in the urbanized area.

Figure 4 shows the taxonomic composition at the level of plant genera of the honey samples from the three different ecological areas of Kazakhstan.

DNA metabarcoding with the *ITS2* primers showed that, among the most frequently occurring plant genera, the following predominated in the intensive farming area: *Sinapis* (54%), *Helianthus* (14%), *Picris *(6%), *Fagopyrum* (5%), and *Melilotus* (4%). According to the data obtained with the *rbcL *primers, the predominant species were *Helianthus* (19%), *Fagopyrum *(16%), *Picris* (12%), *Linaria* (11%), and *Brassica* (9%).

In the reserved area, the most common, according to the *ITS2* data, were *Melilotus* (26%), *Helianthus* (21%), *Brassica* (17%), *Berteroa *(6%), *Filipendula* (5%), and *Onobrychis* (5%). According to the *rbcL *data, the dominant plant genera in the reserved area were *Helianthus *(22%),* Melilotus* (19%), *Brassica* (15%), *Solidago* (12%), *Erodium *(7%), and *Lonicera* (6%).

In the urbanized area, according to the data obtained with the *ITS2 *marker, pollen most often belonged to the following genera:* Raphanus* (17%), *Helianthus *(16%), *Reseda* (12%), *Trifolium* (9%), and *Melilotus *(9%). The *rbcL* marker showed that, in the urbanized area, the dominant plant taxa were *Cucumis* (25%), *Melilotus* (11%), *Helianthus* (9%), *Gossypium *(9%), *Peganum *(8%), and *Malva* (6%).

The most frequently occurring plant species for both markers are shown in Appendix A. As can be seen from the table, the following plant species predominated in the intensive farming area: *Sinapis alba* (72%), *Helianthus divaricatus* (20%), and *Fagopyrum esculentum* (16%). In the reserved area, the most common were *Helianthus divaricatus* (24%), *Filipendula vulgaris* (20%), *Melilotus albus* (20%), *Echium vulgare* (17%), and *Melilotus officinalis* (16%). The dominant plant species in the urbanized area were *Raphanus sativus* (31%), *Trifolium pratense *(16%), and *Cucumis melo* (15%). All identified species are given in Appendix A.

### 3.3. Differences between Foraging Preferences of Bees in Different Ecological Zones: Alpha and Beta Diversity

High alpha diversity was observed in the reserved and intensive farming areas. In the urbanized area, the diversity based on the total number of OTUs was lower, as confirmed by both markers (Figure 5).

As can be seen from Figure 5, the alpha diversity was significantly lower in the urbanized area than in the others (*p* value = 0.00001). The *rbcL* marker showed greater species diversity within groups.

To assess the beta diversity between honey samples from different zones, principal component analysis (PCoA) based on the OTU-level Bray–Curtis distances was used (Figure 6). The PCA analysis confirmed the significant differences between the three studied ecological zones (Bonferroni *p* ≤ 0.05).

### 3.4. Comparison of Metabarcoding and Melissopalynology Data

A comparison of the agreement in identifying the dominant plant species across the three analyses is presented in a Venn diagram (Figure 7).

As can be seen from Figure 7, of the most common plant species, the greatest agreement was observed between the *rbcL* marker and palynological analysis, while the *ITS2* marker revealed almost the same number of matches with both *rbcL* and palynology. Meanwhile, only two species, *Trifolium pretense* and *Brassica nigra*, were identified simultaneously in the three analyses. Species such as *Convolvulus arvensis*, *Melilotus officinalis*, *Echium vulgare*, *Brassica rapa*, *Helianthus divaricatus*, and *Onobrychis viciifolia* were not only identified by the different methods but also occurred in different ecological areas.

## 4. Discussion

To more thoroughly determine the botanical composition of the honey, along with traditional melissopalynological analysis, plant DNA metabarcoding was used at two loci: *ITS2* and *rbcL*. Previous studies have reported that the *ITS2* marker was superior to *rbcL* in identifying plant taxa, and errors in identification at the genus level were low [31]. Even with the identity rates reaching 100%, Bell et al. obtained fewer high-quality sequence reads from *rbcL* than *ITS2*. The authors attributed this to the fact that the *rbcL* fragment is longer and has less overlap between forward and reverse reads, resulting in more reads being removed during filtering [37]. At the same time, Carneiro et al., using an *ITS2* dataset, found fewer plant families in comparison with the *rbcL* [44]. To obtain better results when assessing the biodiversity of the honey’s plant composition, several authors recommend using multiple loci [31,36,44].

In our study, the higher number of plant taxa in honey was obtained using the *rbcL* marker (326 taxa), while 233 taxa were identified using *ITS2*, and 219 taxa were identified using melissopalynological analysis. When comparing the three analyses considering the most represented plants, most matches were between the *rbcL* marker and the melissopalynological analysis. The *ITS2* marker revealed almost the same number of matches with *rbcL* and melissopalynology, but this marker missed or underestimated some relevant genera (e.g., *Sinapis*). Only two species, *Trifolium pretense* and *Brassica nigra*, were identified by all three analyses. Species such as *Convolvulus arvensis*, *Melilotus officinalis*, *Echium vulgare*, *Brassica rapa*, *Helianthus divaricatus*, and *Onobrychis viciifolia* were not only identified by different methods but also occurred in different ecological areas.

Despite the several advantages of NGS, we also used the traditional method of melissopalynology to determine the plant composition of the honey. The studies by Laha et al. showed that palynological data revealed many plant species that could not be identified using NGS. The authors attributed this to the incompleteness of the plant databases [10]. Smart et al. noted that although DNA sequencing analysis identified a more significant number of taxa, this method was limited by the availability of relevant sequences for comparison in reference databases. Some taxa were incorrect and indicated gaps in the database within the phylogenetic lineage [33]. Aligning local species sequences with a local reference database instead of a global database increased the accuracy for *ITS2* from 63% to 75% and for *rbcL* from 46% to 58% [45]. The results of our research indicate that plant taxa databases do not always contain information about local plants. Therefore, the reliability was not always sufficient at the species level, especially with the *ITS2* primers. More plant taxa were detected using the *rbcL* marker, especially at the species level.

The melissopalynological data more accurately determined the plant taxa at the species level, especially in the reserved area and the intensive farming area. To determine the plant taxa, the Atlas of Pollen Grains was used, based on plants growing in a natural area similar in climatic and geographical conditions to the regions of Northern and Central Kazakhstan. However, the DNA metabarcoding of the honey made it possible to discover more plant taxa, especially in the urbanized area, as there is an abundance of plants growing in the country’s southern regions that are not included in the Pollen Plant Atlas. In the urbanized area, some specific plants were more accurately identified using metabarcoding: *Peganum harmala*, *Cucumis melo* subsp. *Melo*, *Glycine max*, *Nitraria sphaerocarpa*, and *Saussurea elegans*.

It should be noted that, in both methodologies, there were many pollen types not identified at the species level.

The landscape class (rural, suburban, and urban) explains the spatial variation in the composition of plants fed by honey bees but not in the taxa richness [46]. In our case, the highest alpha diversity was observed in the reserved and intensive farming areas. The urbanized area had less diversity based on the total number of OTUs than the others (*p* value = 0.00001).

The predominant genera in the intensive farming area were *Sinapis* (54%), *Helianthus* (14%), *Linaria* (11%), *Brassica* (9%), *Picris* (6%), *Fagopyrum* (5%), and *Melilotus* (4%). These data were concordant for both markers, with slight deviations in percentage, and with the data of the melissopalynological analysis. In this area, the apiaries are located near fields with monoculture plants: buckwheat, mustard, and sunflower. Previous studies have reported that bees prefer these plants during flowering in monoculture zones. However, after this period, there is a lack of food, which leads to a deterioration in the nutrition of bees, their productivity, and their health [5,27]. In such cases, an essential factor is the correct management of the apiary, aimed at finding available food resources. In our case, the apiaries migrated from monoculture fields to river floodplains. Therefore, in addition to monoculture plants, an abundance of other plants was observed in the intensive farming area.

In the reserved area, the stationary apiary was surrounded by deciduous and coniferous forests. It has been observed that, in forested landscapes with high forest cover, bees are limited in pollen, and they more often forage for food in meadows and arable lands [21]. In this area, the beekeeper planted plants around the apiary, including those brought from other places to feed the bees. The research results show that the bees used the available plant species around the apiary. In the reserved area, the following plant genera prevailed: *Melilotus* (26%), *Helianthus* (22%), *Brassica* (17%), *Solidago* (12%), *Erodium* (7%), *Lonicera* (6%), *Berteroa* (6%), *Filipendula* (5%), and *Onobrychis* (5%). The biodiversity of the bee foraging plants was greatest in the reserved area. This may have been due to the fact that the beekeeper’s management allowed the bees to use additional available resources near the apiary.

According to the metabarcoding data, in the urbanized area, the following plant genera predominated: *Cucumis* (25%), *Raphanus* (17%), *Helianthus* (16%), *Reseda* (12%), *Trifolium* (9%), *Melilotus* (9%), *Gossypium* (9%), *Peganum* (8%), and *Malva* (6%). At the same time, a large share was accounted for by garden plants, such as radish (*Raphanus sativus*) and cucumber (*Cucumis hystrix*), as well as agricultural plants, such as soybean (*Glycine max*) and melon (*Cucumis melo*).

## 5. Conclusions

The study of bee foraging in different ecological areas is necessary to understand the foraging preferences of bees and for the development of measures to preserve the bee population. Our research showed that bee nutrition varies across different ecological zones: intensive farming, protected areas, and urban areas. The human management of apiaries has an essential influence on the feeding of bees. This made it possible to diversify the monotonous diet of bees in the intensive farming area and provide food for the bees in the forest in the reserved area. An urbanized environment additionally provides crops for bee nutrition and can serve as a strategy to preserve bee populations.

To determine the botanical composition of honey, it is advisable to couple melissopalynology and metabarcoding. The melissopalynological method was more accurate in identifying plant taxa, especially at the species level, but it relies on the availability of a local reference atlas. However, honey DNA metabarcoding revealed a higher number of plant taxa. Specifically, more plant taxa were detected using the *rbcL* marker than *ITS2*, especially at the species level. The imperfection of the databases, particularly for native plants, makes accurate identification using metabarcoding sometimes difficult. Therefore, further research and the expansion of the plant database are necessary.

The study of bee foraging using DNA metabarcoding in three ecological areas was carried out for the first time in Kazakhstan. Our findings may have important implications for the development of beekeeping in the study area.

## Figures and Tables

**Figure 1 insects-15-00674-f001:**
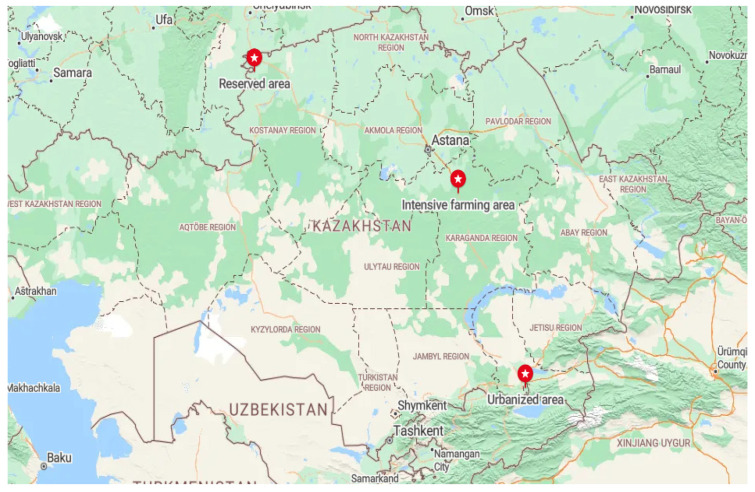
Map showing different sites for collection honey samples from ecological zones of Kazakhstan.

**Figure 2 insects-15-00674-f002:**
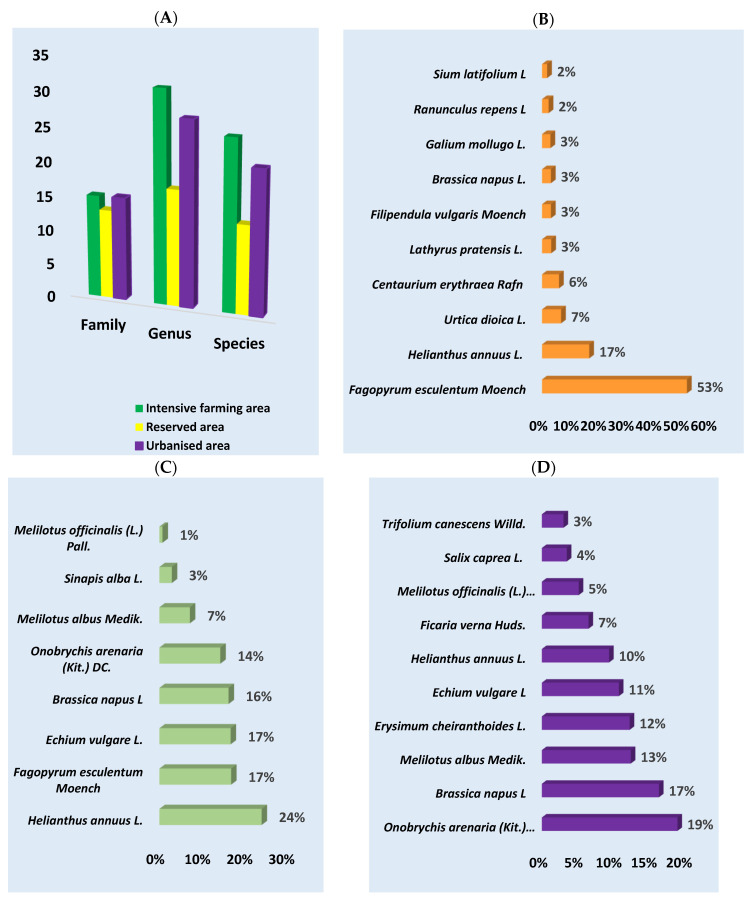
The plant composition of honey from different ecological areas was determined using melissopalynology. (**A**) Number of families, genera, and species that were identified in the samples for each region studied; (**B**) intensive farming area; (**C**) reserved area; (**D**) urbanized area (in percent).

**Figure 3 insects-15-00674-f003:**
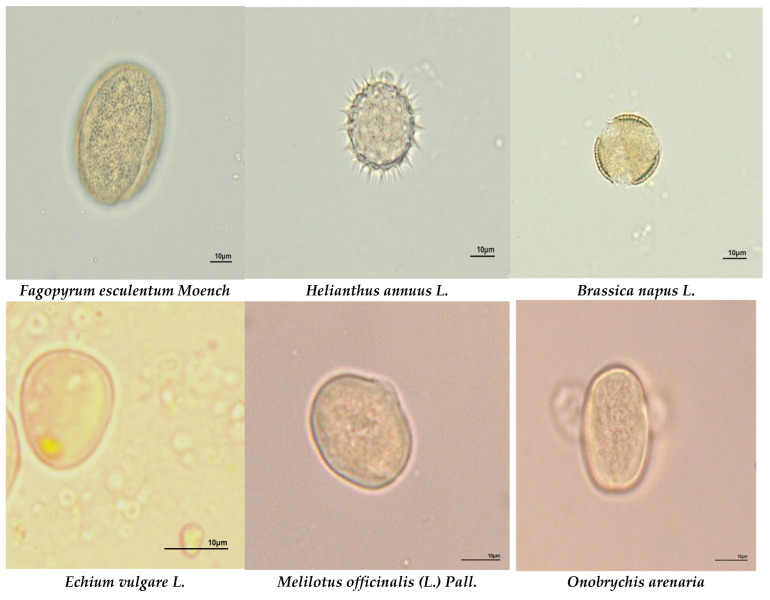
The most common plant species in honey samples from different ecological areas. Microscope magnification ×400.

**Figure 4 insects-15-00674-f004:**
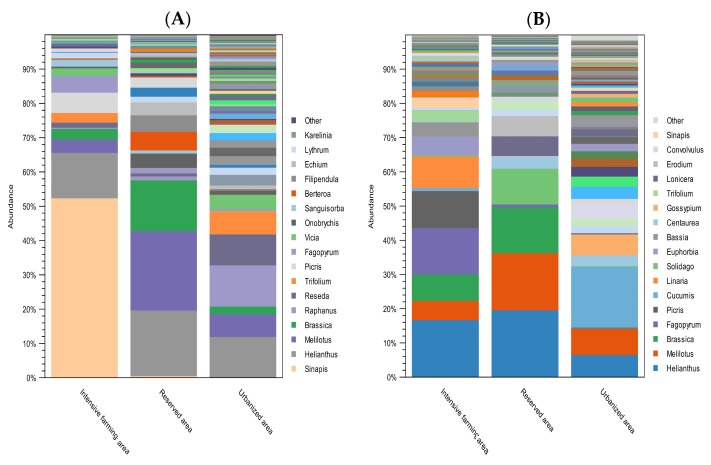
Relative abundance of the most common plant genera found in honey from different ecological areas. (**A**) Plant composition of honey with *ITS2*; (**B**) plant composition of honey with *rbcL*. The abundance of taxa is indicated as a percentage of reads. The height of the bars represents the relative frequency of each plant genus in samples from each study region. Colors are used to differentiate each plant genus. The legend indicates the 17 most abundant plant genera.

**Figure 5 insects-15-00674-f005:**
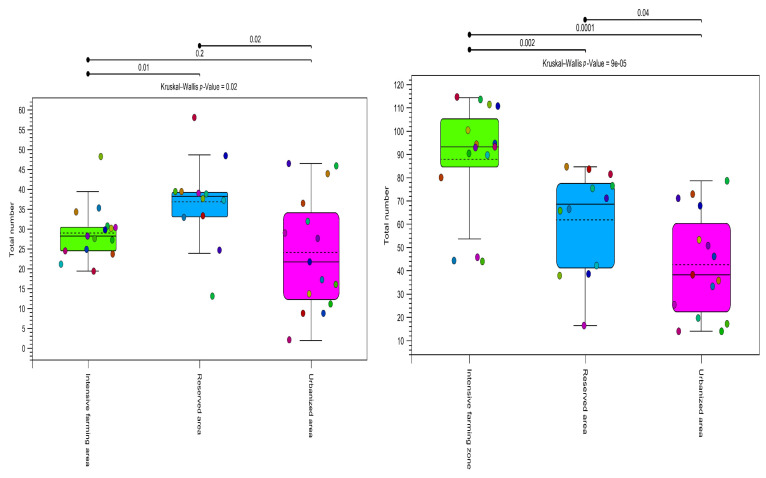
Alpha diversity (total number). Estimates of plant community diversity metrics detected in honey samples using OTU assignments obtained using both metabarcoding markers (*ITS2* on the left, *rbcl* on the right). The colors indicate the three types of ecological areas (intensive farming area, reserved area, and urbanized area). Alpha diversity shows differentiation between all zone types (*p* > 0.05) for all methods used in this study. The results of the Kruskal–Wallis test are indicated in the upper corner of each graph.

**Figure 6 insects-15-00674-f006:**
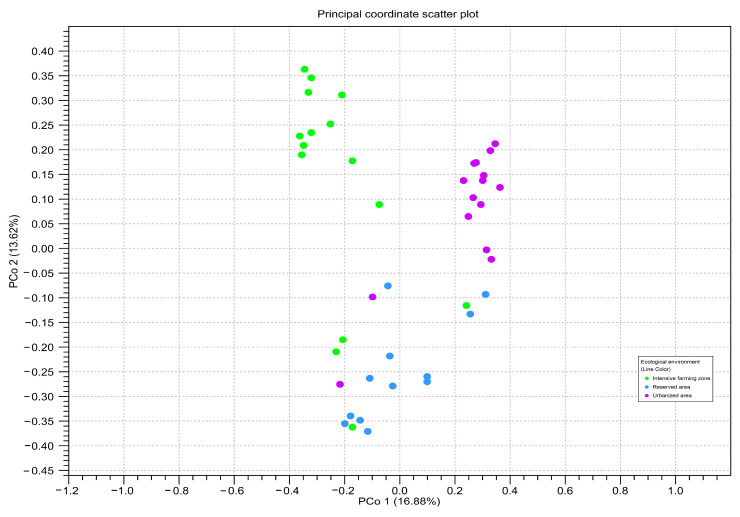
Principal component analysis of read counts for amplicon sequence variants belonging to different ecological areas, calculated using two genetic markers: point = *ITS2*; triangle = *rbcL*. The color of the figure and on the subfigures outline indicates honey samples from different ecological areas: green—intensive farming area; blue—reserved area; pink—urbanized area. The axes represent the first and second principal components, the percentage deviation of which is indicated in parentheses.

**Figure 7 insects-15-00674-f007:**
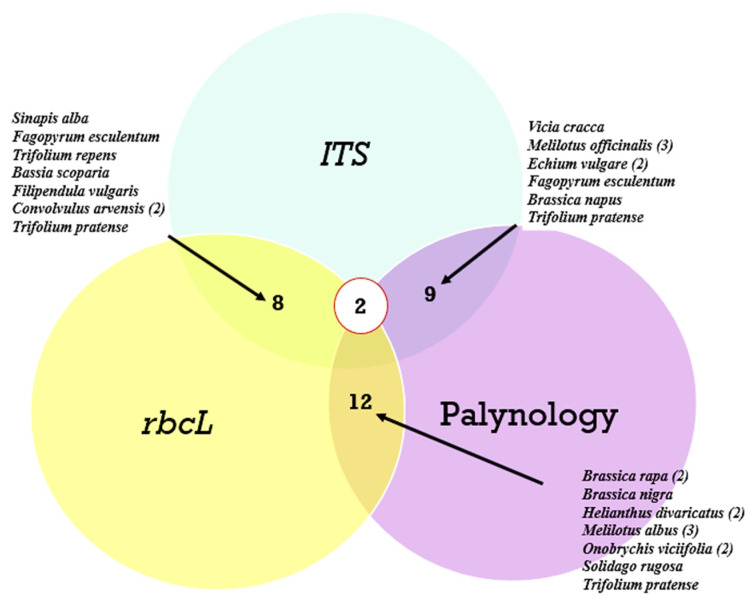
Venn diagrams showing the number of matches by species of the most common plants from all honey samples. Legends indicate the plant species. The number of matches for one species is indicated in parentheses.

**Table 1 insects-15-00674-t001:** Primers and amplification modes.

Target Region	Primer Name	Nucleotide Sequence	PCR Protocol and Product Length in Base Pairs	Source
*rbcL*	*rbcLaf* + adaptor	**TCGTCGGCAGCGTCAGATGTGTATAAGAGACAG**ATGTCACCACAAACAGAGACTAAAGC	Thermal cycling conditions were 95 °C for 2 min; 95 °C for 30 s, 50 °C for 1 min 30 s, 72 °C for 40 s (35 cycles); 72 °C for 5 min, 30 °C for 10 s. ~704 bp.	[11]
	*rbcLr*506 + adaptor	**GTCTCGTGGGCTCGGAGATGTGTATAAGAGACAG**AGGGGACGACCATACTTGTTCA		
*ITS2*	*ITS2* S2F	**TCGTCGGCAGCGTCAGATGTGTATAAGAGACAG**ATGCGATACTTGGTGTGAAT	Thermal cycling conditions were 94 °C 5 min; 94 °C 30 s, 56 °C 30 s, 72 °C 45 s, 40 cycles; 72 °C 10 min. ~350–500 (size range) in bp.	[41]
	*ITS2* S3R	**GTCTCGTGGGCTCGGAGATGTGTATAAGAGACAG**GACGCTTCTCCAGACTACAAT		

Note: the Illumina adapter area is highlighted in bold.

## Data Availability

The data presented in this study are publicly available at the NCBI Sequence Read Archive (Bioproject ID: PRJNA1126018). Other data are contained within the article and Appendix A.

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
