# Peer review of "Foraging of Honeybees from Different Ecological Areas Determined through Melissopalynological Analysis and DNA Metabarcoding"

_insects, 2024, doi:10.3390/insects15090674_

Round 1

Reviewer 1 Report

Comments and Suggestions for Authors

Simple Summary

Line 26: „bee feeding“ should be replaced with „bee diet“.

Line 27: How the agricultural plants serve as primary food for bees in urban areas?

Line 27: „primary food“should be replaced with „primary food sources“.

Line 27: „Feeding of bees“ should be replaced with „ Bee foraging“.

Abstract

In line 43 you wrote „by three analyses“, but you specified only two analyses (melissopalynological analysis and DNA metabarcoding) in line 34. I suggest authors either introduce three analyses or avoid mentioning "three analyses" in line 43.

1. Introduction

Line 60: „Bee feeding“ should be replaced with „Bee foraging“.

Line 61: Please delete: „obtained from plants“ as it is redundant.

Line 100: „Feeding preferences“ should be replaced with „foraging preferences“.

2. Materials and Methods

I think the title of subsection 2. 3. should be renamed to „Metabarcoding“, as metabarcoding is an important part of the work and you have „Metabarcoding“ as a subsection within the „3. Results“ section.

If possible, avoid small subtitles „Preparation of a smear.“ (Line 192), „Light microscopy.“ (Line 202) and „Study of the pollen spectrum.“ (Line 204).

3. Results

Table 1: Markers (target regions) ITS2 and rbcL should be added (in a separate column or the column with the primer names).

Table 1: In the title of the third column I would replace „Amplification mode“ with „PCR protocol“ and „nucleotide pairs“ with „base pairs (bp)“.

Table 1: In the fourth column, instead of full references, leave only [11] and [41].

Lines 351-352: „the most common genus were“ might be replaced with „pollen most often belonged to the following genera:“

Discussion

Line 462: I would replace „taxa“ with „genera“.

Line 464: „plants for feeding bees“ should be replaced with „bee foraging plants“.

Conclusion:

Line 476: „Feeding preferences“ should be replaced with „foraging preferences“.

Line 493: „bee feeding“ should be replaced with „bee forage“.

Comments on the Quality of English Language

I noticed mostly terminological errors (which I pointed out in the "Comments and Suggestions for Authors
"). 

Reviewer 2 Report

Comments and Suggestions for Authors

line 91-99: This whole paragraph is not needed or can be shortened or moved to discussion becasue this study did not investigate the mechanisms under foraging preferences.

line 100: Be cautious when using the term "preference" in this study, as it lacks a survey of the plants growing in each area. Without that information, authors can not conclude that bees forage based on preference rather than simply due to plants abundance. Please double check with this issue in other paragraphs as well.

line 176: Were the honey samples taken from individual hives at different apiaries, or were they a mix of hives from different individual apiaries? Clarifying this would help demonstrate the representativeness of your samples.

line 295-297: This information is repeated as in the previous paragraphs.

line 311: It would be helpful to have a detailed scale for the bars or to show numbers directly on the bars.

line 363-365: this should be mentioned in the methods section.

Overall, this study uses different methods to survey bee hive food diversity across three different areas, reflecting their foraging food composition. The authors should focus more on discussing the differences between these identification methods and the results of survey itself. Besides above comments, the discussion and conclusions need to be reorganized and rephrased becasue of following reasons:

line 444-472: Some of the numbers(results) are repeated presented and should be removed. These paragraphs lack sufficient discussion and results explanation, so the function of these paragraphs is unclear.

line 478, there is no evidence or data to support the statement that "...it feeding bees depends on the availability of plants, the surrounding landscape, and climatic and geographical conditions...". To draw such a conclusion, additional data would be needed, such as how honey nutrient composition and water% of the sampled honey. For assessing foraging based on climate differences, detailed weather data from the two areas would be required. Moreover, since this study only used samples from one-time point, it lacks the data needed to support how climate might change feeding composition in the same area at different timeframes of the year.

Reviewer 3 Report

Comments and Suggestions for Authors

The document is a novel and interesting contribution, but I suggest some minor modifications:

• In the introduction I suggest adding a paragraph on the context of beekeeping in Kazakhstan, and briefly commenting on the previous level of knowledge of bee flora in its different regions. In addition, because lists of species will be compared, it is necessary for the reader to know if the number of plant species potentially present in each type of vegetation to be studied is known.

• Delete the last statement “Studying bee foraging is necessary…” (lines 140 to 142), it is redundant. The idea was established in the first paragraph.

• In materials and methods, in the last paragraph of section 2.1, describe more clearly the way in which the samples were obtained. It is not clear to the reader if it was honey harvested by the beekeepers, if it was extracted by the producer, if each sample came from a different hive, all the samples from the region from the same hive, or if samples were obtained directly from the honeycombs. It also does not indicate the date (or dates) on which the samples were obtained, and whether this date coincides with the harvest season in each region (this is necessary to know if the results are similar to those that could be expected from commercial honeys from each of the regions).

• In results, the first statement (A total of 42 samples…. urbanized areas); lines 295 to 297 is repetitive with the methodology, please delete it.

• Lines 297 to 300. The concept of “morphotypes” is confusing; it is not possible that so many different morphotypes were found (morphotype refers to a classification based on observable characteristics of pollen grains). Most likely they refer to the fact that they quantified 11,156 pollen grains, of which 4,055 were identified at the family level, 3,364 at the genus level, 2,784 at the species level and 953 were not identified. A serious error is that in the analysis of the results (including table 2 of the supplementary file), all the morphotypes that failed to be classified at the species level were omitted. It is unlikely that any of those identified at the family level were represented at more than 1%. Similarly, to calculate the percentages, the authors only consider the pollen types (or morphotypes) identified at the species level, not including at least the “other” category.

• It would be desirable that the results of the melissopalynological analysis also indicate the total number of families, genera and species that were identified in the samples for each region studied.

• In Figure 2, image B, the names of the species are incomplete.

• In table 2, of the supplementary file, it is not necessary to repeat the “genus” column, since it is indicated in the next column.

• Figure 3 should include a scale bar for each image, a bar of 10 µ is suggested. Except for very large grains, it is recommended to photograph at 1000x. In the revised version the images are of low quality, probably because the file is light, but the photographs should be improved.

• Lines 330 to 332. How many taxa at the species level were identified for each region?

• In Table 3 of the supplementary file, it is better to list all the identified species, this is the advantage of it being a supplementary file, more information can be placed there that may be relevant to some readers.

• Line 394. Delete the first statement, it is not necessary.

• In the discussion it is necessary to consider the plant diversity present in each of the regions (or areas) studied. Contrast the findings with floristic lists of each area, to compare the total number of species found with those that are present in the places and periods studied. It is also necessary to integrate the consideration that in both methodologies there were many pollen types not determined at the species level.

• Line 470 italics in “Raphanus sativus

• In line 304 and Figures 2C, 2B and Figure 3 the scientific name of Echium vulgare is misspelled, it says “Echium vulagare”.

• In Figure 2D and Figure 3 it says Melilotus officnalis, it should say “Melitotus officinalis”

• In Figure 2D there is no space between arenaria and (Kit).
